# Genetic Diversity and Growth-Promoting Functions of Endophytic Nitrogen-Fixing Bacteria in Apple

**DOI:** 10.3390/plants14081235

**Published:** 2025-04-18

**Authors:** Hongshan Liu, Huan Cheng, Suwen Xu, Donghua Zhang, Jianrong Wu, Zongyan Li, Benzhong Fu, Li Liu

**Affiliations:** 1Key Laboratory of Forest Disaster Early Warning and Control of Yunnan Provincial, Forestry College, Southwest Forestry University, Kunming 650224, China; lhs393393@163.com (H.L.); ch08252025@163.com (H.C.); suesama@163.com (S.X.); fooifoot@126.com (D.Z.); wujianrong63@aliyun.com (J.W.); 2Key Laboratory of Forest Resources Conservation and Utilization in the Southwest Mountains of China, Ministry of Education, Forestry College, Southwest Forestry University, Kunming 650224, China; 3Landscape Architecture and Horticulture Faculty, Southwest Forestry University, Kunming 650224, China; lizyan@swfu.edu.cn; 4Department of Plant Pathology, College of Agronomy, Xinjiang Agricultural University, 311 East Nongda Road, Urumqi 830052, China

**Keywords:** apple tree, endophytic diazotrophs, genetic diversity, nitrogenase activity, growth-promoting

## Abstract

Understanding the dominant populations and biological functions of endophytic nitrogen-fixing bacteria in apple plants is of great significance for the healthy growth management and sustainable development of apple cultivation. In this study, we investigated the community diversity and potential plant growth-promoting abilities of endophytic nitrogen-fixing bacteria in different tissues of apple trees by combining high-throughput sequencing of the *nifH* gene with traditional isolation and cultivation techniques. Sequencing results revealed that the endophytic bacteria were affiliated with 10 phyla, 14 classes, 30 orders, 42 families, and 72 genera. *Rhizobium* was the dominant genus in the roots and twigs, while *Desulfovibrio* dominated the leaf tissues. The diversity and richness of endophytic bacteria in the roots were significantly higher than those in the leaves. Using four types of nitrogen-free media, a total of 138 presumptive endophytic nitrogen-fixing bacterial strains were isolated from roots, leaves, and twigs. These isolates belonged to 32 taxonomic groups spanning 5 phyla, 8 classes, 11 orders, 13 families, and 18 genera. The *nifH* gene was successfully amplified from the representative strains of all 32 groups using specific primers. Nitrogenase activity among the isolates ranged from 26.86 to 982.28 nmol/(h·mL). Some strains also exhibited the ability to secrete indole-3-acetic acid (IAA), solubilize phosphate and potassium, and produce siderophores. Six individual strains and three microbial consortia were tested for their plant growth-promoting effects on apple tissue culture seedlings. All treatments showed growth-promoting effects to varying degrees, with the RD01+RC16 consortium showing the most significant results: plant height, number of leaves, and chlorophyll content were 2.4, 3.3, and 4.2 times higher than those of the control, respectively. These findings demonstrate the rich diversity of endophytic nitrogen-fixing bacteria in apple plants and their promising potential for application in promoting host plant growth.

## 1. Introduction

Apple (*Malus pumila* Mill.), a deciduous tree belonging to the Rosaceae family, *Malus* genus, is one of the most widely cultivated fruit tree species globally [1]. Nitrogen is a crucial factor for apple yield. To achieve high productivity, some apple orchards apply chemical nitrogen fertilizer at rates as high as 600–800 kg per hectare [2,3]. However, excessive application of chemical fertilizers increases production costs, negatively impacts soil health [4], and leads to substantial emissions of the greenhouse gas N_2_O [5]. Therefore, biofertilizers represent a promising green alternative in apple cultivation and management.

Plant growth-promoting bacteria (PGPB), which colonize the rhizosphere, plant surfaces, or internal tissues, promote plant growth and development, and improve crop yield and quality through various mechanisms, such as secreting phytohormones [6], nitrogen fixation, phosphate and potassium solubilization, and enhancing plant stress resistance [7]. Endophytic nitrogen-fixing bacteria are a key category of PGPB. These bacteria can enter plant tissues and convert atmospheric nitrogen into plant-available ammonium nitrogen without forming root nodules [8,9], thereby providing nitrogen nutrition and promoting plant growth and development. In 1988, Cavalcante and Dobereiner pioneered the field of endophytic nitrogen fixation by isolating the endophytic nitrogen-fixing bacterium *Gluconacetobacter diazotrophicus* from sugarcane tissue [10]. Consequently, the focus on diazotrophic systems has expanded beyond gramineous crops to include bananas [11], coffee trees [12], pineapples [13], strawberries [14], medicinal plants [15], and more. However, research and development on endophytic nitrogen-fixing bacteria in woody plants remain relatively limited. Endophytic diazotrophic microorganisms have shown significant growth-promoting effects in various crops, including rice [16], wheat [17], maize [18], and other agricultural plants [19]. In some cases, the growth-promoting function of endophytic nitrogen-fixing bacteria even surpasses that of chemical nitrogen fertilizers [20]. Moreover, increasing evidence suggests that endophytic nitrogen-fixing bacteria can indeed assist host plants in nitrogen fixation. For example, Boddey et al. found that inoculation with endophytic nitrogen-fixing bacteria could satisfy 60–80% of the nitrogen requirements for sugarcane growth [21]. Inoculation with endophytic nitrogen-fixing bacteria enabled lodgepole pine to acquire 23–53% of its nitrogen directly from atmospheric nitrogen fixation [22]. 

Moreover, many endophytic nitrogen-fixing bacteria have demonstrated strong antagonistic activity against plant pathogens [23] and the ability to induce plant disease resistance [24]. Therefore, considering their combined functions in nitrogen fixation, plant growth promotion, and disease suppression, endophytic nitrogen-fixing bacteria show great potential for development as multifunctional biofertilizers.

## 2. Results

### 2.1. Diversity and Community Structure of Endophytic Nitrogen-Fixing Bacteria in Apple Plants

#### 2.1.1. Alpha Diversity of Endophytic Nitrogen-Fixing Bacteria in Apple Plants

High-throughput sequencing of the *nifH* gene was performed on three tissue types (root, twig, and leaf) of apple plants, yielding 163,168, 59,522, and 32,364 raw reads, respectively. Clustering of operational taxonomic units (OTUs) was performed at a 97% sequence similarity level, resulting in 4660, 883, and 291 OTUs, respectively. The Good’s coverage values ranged from 98% to 100% (Table 1), indicating sufficient sequencing depth to adequately assess the community composition of each sample.

Alpha diversity analysis was conducted using group comparisons and significant difference tests to compare the diversity and richness of endophytic nitrogen-fixing bacteria in different tissues. The Shannon and Simpson indices showed that the diversity of root endophytic nitrogen-fixing bacteria was significantly higher than that of leaf endophytic nitrogen-fixing bacteria (*p* < 0.01). The Chao1 and ACE indices showed that the richness of root endophytic nitrogen-fixing bacteria was significantly higher than that of leaf endophytic nitrogen-fixing bacteria (*p* < 0.05) (Table 1). There was no significant difference in the diversity and richness of twig endophytic nitrogen-fixing bacteria compared to root and leaf samples.

#### 2.1.2. Beta Diversity of Endophytic Nitrogen-Fixing Bacteria in Apple Plants

Principal Coordinate Analysis (PCoA) was performed on the weighted Bray–Curtis dissimilarity matrix to analyze the differences in community structure among samples (Figure 1). The results showed that the first and second principal coordinates explained 33.97% and 21.47% of the variability in the data, respectively. The endophytic nitrogen-fixing bacterial communities in apple twigs (AT) and leaves (AL) exhibited relatively similar structures. In contrast, the root endophytic nitrogen-fixing bacterial communities (AR) were clearly separated from those in twigs and leaves, indicating that plant tissue type influenced the community structure and diversity of endophytic nitrogen-fixing bacteria.

#### 2.1.3. Community Composition of Endophytic Nitrogen-Fixing Bacteria in Apple Plants

Clustering analysis of sample sequences at a 97% similarity level revealed that endophytic nitrogen-fixing bacteria in apple roots could be classified into 10 phyla, 14 classes, 30 orders, 42 families, and 72 genera. Endophytic nitrogen-fixing bacteria in twigs were classified into 3 phyla, 5 classes, 10 orders, 14 families, and 15 genera. Endophytic nitrogen-fixing bacteria in leaves were classified into 1 phylum, 3 classes, 5 orders, 7 families, and 6 genera.

At the phylum level (Figure 2), root endophytic nitrogen-fixing bacteria were classified into Pseudomonadota, Actinomycetota, Bacillota, and other phyla with relative abundances <0.1%. Planctomycetota, Bacillota, Deinococcota, Acidobacteriota, Nitrospirota, Bacteroidota, and Verrucomicrobiota were unique to the root samples. Twig endophytic nitrogen-fixing bacteria belonged to the phyla Pseudomonadota, Actinomycetota, and Cyanobacteriota. Leaf endophytic nitrogen-fixing bacteria were only classified into the phylum Pseudomonadota. A significant proportion of unclassified sequences was also observed in all tissue types.

At the genus level (Figure 2), the composition of endophytic nitrogen-fixing bacterial communities differed substantially among tissues. Root endophytic nitrogen-fixing bacteria were classified into 72 genera. The top 10 most abundant genera were *Rhizobium* (12.15%), *Marinobacterium* (0.95%), *Bradyrhizobium* (0.86%), *Agrobacterium* (0.35%), *Pseudolabrys* (0.24%), *Pseudomonas* (0.15%), *Paenibacillus* (0.13%), *Mesorhizobium* (0.08%), *Actinoplanes* (0.06%), and *Mycolicibacterium* (0.05%).

Twig endophytic nitrogen-fixing bacteria were classified into 15 genera. The top 10 most abundant genera were *Rhizobium* (0.042%), *Bradyrhizobium* (0.034%), *Mycolicibacterium* (0.018%), *Actinocatenispora* (0.005%), *Arthrobacter* (0.005%), *Dokdonella* (0.005%), *Pseudolabrys* (0.003%), *Mesorhizobium* (0.003%), *Sandaracinus* (0.003%), and *Afipia* (0.002%). No genera were unique to the twig samples.

In the leaf tissues, endophytic nitrogen-fixing bacteria were identified from only six genera. Their relative abundances, in descending order, were *Desulfovibrio* (0.09%), *Methylocystis* (0.046%), *Brevundimonas* (0.025%), *Ancylobacter* (0.015%), *Novosphingobium* (0.009%), and *Rhodovulum* (0.006%).

The heatmap displays the average relative abundance of each OTU within each sample type. AR = apple root, AT = apple twig, AL = apple leaf. OTUs were clustered based on their abundance patterns.

### 2.2. Isolation and Identification of Endophytic Nitrogen-Fixing Bacteria in Apple

A total of 138 endophytic nitrogen-fixing bacterial strains were isolated from different tissues of apple plants using four different nitrogen-free media. Among these isolates, 64 were derived from roots, 34 from twigs, and 40 from leaves (Figure 3). Based on 16S rRNA gene sequence comparisons, these isolates were identified as belonging to 5 phyla, 8 classes, 11 orders, 13 families, and 18 genera, comprising 32 taxonomic groups (Table 2). Except for strains ZD-05, LB-03, LD-11, LB-01, RC-04, and ZC-10, the remaining isolates exhibited >99% sequence similarity to known species. The isolation frequencies, in descending order, were as follows: *Agrobacterium* (23.20%), *Novosphingobium* (15.95%), *Sphingomonas* (13.04%), *Sphingobium* (10.14%), *Methylobacterium* (8.0%), *Nocardioides* (5.80%), *Brucella* (5.80%), *Curtobacterium* (4.35%), *Starkeya* (2.90%), *Ancylobacter* (2.17%), *Bacillus* (2.17%), *Pseudomonas* (1.45%), *Rhizobium* (1.45%), *Deinococcus* (0.72%), *Microbacterium* (0.72%), *Pseudorhodoplanes* (0.72%), *Rhodococcus* (0.72%), and *Sphingobacterium* (0.72%).

### 2.3. Nitrogenase Activity of Endophytic Nitrogen-Fixing Bacteria

Nitrogenase activity was measured in 32 representative strains using the acetylene reduction assay (Table 3). Nitrogenase activity in root-derived endophytic nitrogen-fixing bacteria ranged from 27.1 to 425.3 nmol/(h·mL), with the highest activity observed in strain RD-01 (*N. barchaimii*). Twig-derived endophytic nitrogen-fixing bacteria exhibited nitrogenase activity ranging from 74.95 to 982.28 nmol/(h·mL), with the highest activity observed in strain ZC-06 (*R. qingshengii*). Leaf-derived endophytic nitrogen-fixing bacteria showed nitrogenase activity ranging from 26.86 to 543.95 nmol/(h·mL), with the highest activity observed in strain LC-01 (*S. multivorum*). Therefore, these three strains, derived from different tissue types (root, twig, and leaf) and exhibiting the highest nitrogenase activity, were selected for subsequent growth promotion experiments.

### 2.4. Assessment of Other Growth-Promoting Potential of Apple Endophytic Nitrogen-Fixing Bacteria

The 32 representative strains (each one strain from the 32 taxonomic groups in Section 2.2) were further screened for their ability to solubilize phosphorus, solubilize potassium, produce siderophores, and produce IAA (Figure 4). The results showed that five strains exhibited the ability to solubilize inorganic phosphorus, with solubilization capacities ranging from 383.37 ± 0.028 μg/mL to 139.01 ± 0.002 μg/mL. Thirteen strains were able to solubilize organic phosphorus, with solubilization capacities ranging from 368.26 ± 0.015 μg/mL to 24.00 ± 0.120 μg/mL. Eight strains exhibited potassium solubilization ability, with clear zone diameters (D/d) ranging from 2.50 ± 0.012 to 0.89 ± 0.005. Eight strains produced siderophores, with clear zone diameters (D/d) ranging from 7.86 ± 0.052 to 1.40 ± 0.310. Thirty strains were able to produce IAA, with IAA production ranging from 39.00 ± 0.045 μg/mL to 5.96 ± 0.022 μg/mL.

Among these, strain RC-16 (*N. resinovorum*) exhibited all four growth-promoting traits (inorganic phosphorus solubilization, IAA production, siderophore production, and potassium solubilization). Strain ZC-11 (*S. yunnanensis*) exhibited three growth-promoting traits (organic phosphorus solubilization, IAA production, and siderophore production). Strain LD-06 (*S. yabuuchiae*) exhibited all four growth-promoting traits (organic phosphorus solubilization, IAA production, siderophore production, and potassium solubilization). Therefore, these three strains, derived from different tissue types and exhibiting multiple growth-promoting potential, were selected for subsequent growth promotion experiments.

### 2.5. Growth-Promoting Effects of Apple Endophytic Nitrogen-Fixing Bacteria

Apple tissue culture seedlings were inoculated with single strains (RD-01, ZC-06, LC-01, RC-16, ZC-11, and LD-06) and combined strains (RD-01 + RC-16, ZC-11 + ZC-06, and LC-01 + LD-06) at equal proportions. These strains were selected based on their tissue origin (root, twig, and leaf), nitrogenase activity (RD-01, ZC-06, LC-01), and multiple growth-promoting potential (RC-16, ZC-11, LD-06) (Figure 5).

Compared to the control, inoculation with the single strain RD-01 resulted in the most significant increase in plant height, number of leaves, and chlorophyll content, reaching 2.0, 3.8, and 3.2 times that of the control, respectively. Strain ZC-11, derived from twigs, exhibited the most significant promotion of root length, reaching 3.41 times that of the control (Table 4).

Among the combined inoculations, RD-01 + RC-16 showed the most significant increase in plant height, number of leaves, and chlorophyll content, reaching 2.4, 3.3, and 4.2 times that of the control, respectively. Moreover, the RD-01 + RC-16 combination significantly outperformed the single strain RD-01 (Table 4).

## 3. Discussion

### 3.1. Diversity of Endophytic Nitrogen-Fixing Bacteria in Apple Plants

Endophytic nitrogen-fixing bacteria colonize within plants, and their community structure and diversity are influenced by multiple factors such as plant species, growth environment, growth stage, and tissue type. In this study, we analyzed the diversity of endophytic nitrogen-fixing bacteria in apple using high-throughput sequencing technology. Beta diversity analysis revealed that the community structure of endophytic nitrogen-fixing bacteria in apple roots was significantly different from that in twigs and leaves. Alpha diversity analysis showed that the diversity and richness of endophytic nitrogen-fixing bacteria in apple roots were significantly higher than those in leaves. These results are similar to those of Zhang et al. [25], who used high-throughput amplification sequencing of the *nifH* gene and found that the Chao1 index of endophytic nitrogen-fixing bacteria in the leaf tissue of pioneer plants on the Qinghai-Tibet Plateau was higher, but the Simpson and Shannon diversity indices were significantly lower than those in the roots (*p* < 0.05). It is now generally believed that plant endophytes originate from the soil [26]. Zhang et al. [27] confirmed through the microbial source tracking tool FEAST that 61% of the endophytes in maize roots originated from the soil, while only 24% in the stems and leaves originated from the soil. The results of this study confirm the speculation about the origin of plant endophytes to a certain extent.

### 3.2. Community Composition of Endophytic Nitrogen-Fixing Bacterial in Apple Plants

Analysis of the community composition of apple endophytic nitrogen-fixing bacteria in this study revealed that the dominant phylum of apple endophytic nitrogen-fixing bacteria is Pseudomonadota, followed by Actinomycetota. Analysis of apple endophytic bacterial communities has shown that the abundance of apple endophytic bacteria decreases in the order of Pseudomonadota, Bacillota, Actinomycetota, Bacteroidota, and Fusobacteriota [28]. The community composition pattern of apple endophytic nitrogen-fixing bacteria at the phylum level is consistent with the pattern of apple endophytic bacteria. The abundance of endophytic nitrogen-fixing bacteria belonging to Pseudomonadota in apple tissues is as high as 15% in the roots but less than 1% in the twigs and leaves. Pseudomonadota are gram-negative bacteria and are eutrophic bacteria, which are positively correlated with nutrient content [29]. This may be the reason why Pseudomonadota account for a larger proportion in the roots. Research on endophytic nitrogen-fixing bacteria is currently focused on some gramineous crops. The reported endophytic nitrogen-fixing bacteria in crops such as sugarcane [30], maize [31], and rice [32] are also dominated by the Pseudomonadota phylum.

At the genus level, the dominant genera in apple roots and twigs are *Rhizobium*, belonging to α-Pseudomonadota, and the dominant genus in leaves is *Desulfovibrio*, belonging to Thermodesulfobacteriota. Among nitrogen-fixing bacteria, *Rhizobium*, *Frankia*, and other genera are common symbiotic nitrogen-fixing bacteria in plants, and their symbiotic nitrogen fixation efficiency is relatively high [33]. *Rhizobium* has also been identified as an endophytic nitrogen-fixing bacterium in the stems of plants such as maize [31] and poplar [34], although it is not a dominant genus in these hosts. *Desulfovibrio* is an important group of anaerobic bacteria [35], which are common endophytic nitrogen-fixing bacteria in some submerged plants [36]. In addition, this endophytic nitrogen-fixing bacterium is more abundant in the roots of *Panicum virgatum* L. under low nitrogen levels [37]. It is speculated that these may be the reasons why this genus is unique and most abundant in apple leaves.

### 3.3. Isolation of Endophytic Nitrogen-Fixing Bacteria from Apple and Its Nitrogenase Activity

The exploration of endophytic nitrogen-fixing bacteria is gradually developing in woody plants [38,39]. We isolated and identified 18 genera of endophytic nitrogen-fixing bacteria from apple tissues. The highest isolation rate in the roots was *Agrobacterium*, in the twigs was *Novosphingobium*, and in the leaves was *Sphingomonas. Agrobacterium* sp. with nitrogen-fixing function has been reported in the root tissue of Teak (*Tectona grandis*) and can significantly increase the nitrogen content in the leaves of seedlings [40]. *Novosphingobium* sp. N034 has been identified as the keystone of endophytic nitrogen-fixing bacteria in mangrove plants [41], and rice endophytic nitrogen-fixing bacteria also include this genus [42]. The top four most abundant endophytic nitrogen-fixing bacteria in giant reed and switchgrass belong to the genus *Sphingomonas* and have the strongest nitrogenase activity [43].

Studies on the nitrogenase activity of isolated endophytic nitrogen-fixing bacteria have found that the nitrogenase activity of endophytic nitrogen-fixing bacterial strains derived from apple twigs is generally higher than that of roots and leaves. This may be because the low O_2_ level in the xylem provides a favorable environment for biological nitrogen fixation and provides the metabolites necessary for nitrogen exchange [44]. The *R. qingshengii* ZC-06 strain isolated from apple twigs in this study has a nitrogenase activity as high as 982.28 nmol/h mL, which is higher than the nitrogenase activity of endophytic nitrogen-fixing bacteria from pepper [45], vetiver [46], and wild rice [47]. The genus *Rhodococcus* is known for its wide range of multifunctional metabolic compounds [48], but there are very few studies on its nitrogen fixation [49]. This paper reports for the first time that *R. qingshengii* bacteria is a plant endophytic nitrogen-fixing bacterium. However, the contribution of these high nitrogenase activity strains to the nitrogen required for plant growth needs further study.

### 3.4. Growth Promoting Characteristics of Endophytic Nitrogen-Fixing Bacteria in Apple

The endophytic nitrogen-fixing bacteria isolated from different tissues of apple in this study almost all have the ability to produce IAA, with the highest reaching 39.00 ± 0.045 μg/mL. Studies by Inga et al. [50] have shown that nitrogen-fixing bacterial strains account for 66% of apple phyllosphere endophytes, and IAA-producing strains account for 50% of apple phyllosphere endophytes, and there is much overlap between the two, which is basically consistent with the conclusions of this study. Studies by Roberto et al. [51] have found that endophytic bacteria with high IAA production can significantly improve the nitrogen fixation capacity of strains. The availability of elements such as phosphorus and potassium can also strongly affect biological nitrogen fixation [52,53]. Studies have confirmed that co-inoculation of nitrogen-fixing bacteria (*Paenibacillus beijingensis* BJ-18) and phosphorus-solubilizing bacteria (*Paenibacillus* sp. B1) significantly increased the nitrogen content of wheat plants [54]. It can be seen that endophytic nitrogen-fixing bacteria with multifunctional properties can not only improve the combined nitrogen fixation efficiency but also promote plant growth in many ways.

Some plant growth-promoting rhizobacteria (PGPR) have been verified by field experiments to increase the yield, growth, and nutrition of apple trees, such as *Bacillus* and *Microbacterium* [30], *Burkholderia* and *Pseudomonas* [32], *Streptomyces* [55], and other commonly used agricultural rhizospheric growth-promoting bacteria. Endophytic bacteria that promote plant growth are a type of plant growth-promoting bacteria (PGPB) that colonize between plant cells. Compared with rhizospheric growth-promoting bacteria, the exploration and development of endophytic nitrogen-fixing bacteria as biofertilizers are less susceptible to adverse environmental conditions [56], and it is expected to find new potential strains. Therefore, this study started from the perspective of apple endophytic nitrogen-fixing bacteria, screened and inoculated six strains of apple endophytic nitrogen-fixing bacteria, and verified that they have growth-promoting ability for apple tissue culture seedlings. The six strains belong to four genera: *Novosphingobium*, *Sphingobacterium*, *Sphingomonas*, and *Rhodococcus*. Among them, *Novosphingobium* and *Sphingomonas* have been reported as rice endophytic nitrogen-fixing bacteria and have obvious growth-promoting effects on rice [42]. *Sphingobacterium* has also been reported to have nitrogen-fixing function [57] and growth-promoting ability on *Vigna mungo* [58] and tomato [59]. *R. qingshengii* is a gram-positive actinomycete, which has been reported to have nitrogen-fixing ability [49,59] and growth-promoting function on the legume plant *Sulla spinosissima* L. [60]. The apple growth-promoting bacterial genera screened in this study basically have some reports of growth-promoting functions in other plants, and field experiments on apple cultivation can be further carried out.

In the process of apple cultivation, the demand for nitrogen fertilizer is much higher than that of field crops such as wheat and rice [61]. If all apple growth is met by using synthetic nitrogen fertilizers, it will inevitably increase mineralization and reduce soil organic matter [62], which is contrary to the concept of sustainable plant cultivation. Therefore, the development of sustainable bacterial fertilizers based on apple endophytic nitrogen-fixing bacteria is a potential choice to solve this dilemma. This study mainly focused on the community structure and growth-promoting effects of apple endophytic nitrogen-fixing bacteria and did not deeply explore the contribution rate of these endophytic nitrogen-fixing bacteria to the nitrogen required for apple growth. After screening out better growth-promoting strains, the growth-promoting mechanism of the strains can be deeply analyzed.

In this study, a total of 138 endophytic nitrogen-fixing bacterial strains were isolated from apple trees, and their plant growth-promoting effects on apple seedlings were evaluated. These results provide strong data support for the further application and development of endophytic nitrogen-fixing bacteria in apple cultivation. However, further validation under field conditions is still needed.

## 4. Materials and Methods

### 4.1. Site Description and Sampling

The sampling site was located at the apple orchard base of Dingshun Agricultural Technology Co., Ltd. in Tuanjie Township, Kunming City, Yunnan Province, southwestern China (102°48′ E, 25°11′ N), with an average altitude of 2195 m, an average annual rainfall of 800–1200 mm, and an average annual temperature of 13.2 °C. The soil type was classified as red soil with low soil pH, low organic matter content, poor nutrient retention capacity, and prone to compaction. The apple cultivar was ‘German Fuji’ (with M26 rootstock). Samples of roots (AR), twigs (AT), and leaves (AL) were collected from healthy 11-year-old trees. Twig samples consisted of 10–20 randomly selected one-year-old twigs (approximately 0.5–1.2 cm in diameter and 20–40 cm in length). Root samples were a composite of primary and fibrous roots (primary roots were taken from 30 to 50 cm below the root base, and fibrous roots were taken from 5 to 20 cm, with a total root sample weight of approximately 50–100 g). Leaf samples consisted of 20–30 newly developed leaves collected evenly from the upper, middle, and lower layers of the canopy, as well as from the east, south, west, and north aspects. Samples were placed in a foam box with ice packs for preservation and transported back to the laboratory for surface disinfection and subsequent experiments.

### 4.2. DNA Extraction, PCR Amplification, and High-Throughput Sequencing

Healthy apple root, twig, and leaf samples were rinsed with running water for 10 min to remove surface dirt. Samples were further disinfected with 70% ethanol for 60 s, followed by disinfection with 5% sodium hypochlorite for 3 min (roots), 2 min (twigs), or 30 s (leaves). Finally, samples were rinsed four times with sterile water. A 100 μL aliquot of the sterile water from the final rinse was spread onto the NA medium. The absence of colony growth indicated thorough surface disinfection. After complete surface disinfection, leaf and root/twig phloem tissues were dissected and fragmented with scissors for total DNA extraction.

Total DNA was extracted according to the manufacturer’s instructions for the FastStart Essential DNA Green Master DNA (Sangon Biotech, Shanghai, China) Extraction Kit. The *nifH* gene was amplified using the universal primers POLF (5’-TGCGAYCCSAARGCBGACTC-3’) and POLR (5’-ATSGCCATCATYTCRCCGGA-3’) [63]. The PCR reaction mixture (50 μL) contained 2 μL of DNA template, 1 μL of each primer (10 μM), 25 μL of Mix, and 21 μL of ddH2O. The PCR amplification conditions were as follows: 98 °C for 2 min; 30 cycles of 98 °C for 15 s; 55 °C for 30 s; and 72 °C for 30 s; and a final extension at 72 °C for 5 min. After PCR product recovery (Vazyme VAHTSTM DNA Clean Beads), the products were quantified using a fluorescence quantification kit (Quant-iT PicoGreen dsDNA Assay Kit, Sangon Biotech, Shanghai, China). Sequencing was performed on the Illumina Miseq platform at Kunming Hanling Biotechnology Co., Ltd. (Kunming, China).

High-throughput sequencing data were processed using Illumina’s official bcl2fastq software (v2.20.0.422) to remove adapter sequences [64]. Usearch software (v11.0.667) was used for quality control and assembly of paired end reads. Non-redundant sequences were extracted from the optimized sequences of each sample to reduce redundant calculations during the analysis. All non-redundant sequences were merged, and singletons (sequences without replicates) were removed. Non-singleton sequences were clustered into OTUs based on 97% similarity. Chimeras were removed during the clustering process, resulting in representative sequences for each OTU. All optimized sequences were aligned to the OTU representative sequences, and sequences with ≥97% similarity to the representative sequences were assigned to the corresponding OTUs, generating an OTU table. Shannon, Simpson diversity indices, and Chao1 and ACE richness indices were calculated using the “vegan” package (2.6.8). Principal Coordinate Analysis (PCoA) based on the Bray–Curtis dissimilarity matrix was performed using the “gplots” package (3.2.0) to reveal differences in endophytic nitrogen-fixing bacterial community composition and structural trends.

### 4.3. Isolation and Identification of Endophytic Nitrogen-Fixing Bacteria

Approximately 2 g of sterile tissue fragments were homogenized in a sterile mortar with 18 mL of deionized water. After settling for 10 min, the supernatant was serially diluted to 10^−1^, 10^−2^, and 10^−3^. A 100 μL aliquot of each dilution was spread onto four different nitrogen-free media: Jensen, A4, Ash, and Jnfb (Table 5). Each dilution was plated in triplicate and incubated at 28 °C in an inverted position. After visible single colonies developed (approximately 5 days), strains were purified by repeated streaking and stored at −80 °C for further use.

Total DNA was extracted from the isolated strains using the Ezup Column Bacterial Genomic DNA Extraction Kit (Sangon Biotech, Shanghai, China). The 16S rDNA gene was amplified using the universal primers 27F (5’-AGAGTTTGATCCTGGCTCAG-3’) and 1492R (5’-GGTTACCTTGTTACGACTT-3’) [65]. PCR products were analyzed by 1.7% agarose gel electrophoresis and sent to Sangon Biotech (Shanghai, China) for sequencing. The resulting 16S rDNA sequences were compared against known sequences in the EzBioCloud database (www.ezbiocloud.net, accessed on 21 September 2024) [66].

The *nifH* gene of the bacterial isolates was amplified using nested PCR to confirm their nitrogen-fixing potential. The first round of PCR used primers FGPHl9 (5’-TACGGCAARGGTGGNATHG-3’) and PolyR (5’-ATSGCCATCATYTCRCCGGA-3’), and the second round used primers AQER (5’-GACGATGTAGATYTCCTG-3’) and PolyF (5’-TGCGAYCCSAARGCBGACTC-3’) [67].

### 4.4. Nitrogenase Activity Assay

Strains confirmed to possess the *nifH* gene were further assessed for nitrogen-fixing ability using the acetylene reduction assay [68]. A 200 μL aliquot of freshly cultured bacterial suspension was inoculated into 25 mL serum vials containing 5 mL of Jensen nitrogen-free medium and incubated at 28 °C and 180 rpm for 48 h. Ten percent of the gas (2 mL) was withdrawn and replaced with an equal volume of acetylene gas. After further incubation for 24 h, a 200 μL gas sample was taken, and the amount of ethylene produced was measured using a gas chromatograph (GC9790II, Manufacturer: Zhejiang Fuli Analytical Instrument Co., Ltd., Wenling, China). The gas chromatograph settings were: injector 200 °C, detector 230 °C, and column oven 80 °C.

Nitrogenase activity = (Sample C_2_H_4_ peak area × Standard C_2_H_4_ concentration × Serum vial volume)/(Standard C_2_H_4_ peak area × Incubation time × 24.9) [69].

### 4.5. Growth Promotion Assay

Six selected strains were used for single and co-inoculation (1:1 ratio) of apple tissue culture seedlings to determine their plant growth-promoting activity. The strains selected were RD-01 (root-derived, high nitrogenase activity), ZC-06 (twig-derived, high nitrogenase activity), LC-01 (leaf-derived, high nitrogenase activity), RC-16 (root-derived, multiple growth-promoting potential), ZC-11 (twig-derived, multiple growth-promoting potential), and LD-06 (leaf-derived, multiple growth-promoting potential), as well as the co-inoculation treatments RD01 + RC16, ZC11 + ZC06, and LC01 + LD06 (Table 6).

#### 4.5.1. Preparation of Tissue Culture Seedlings

Apple tissue culture seedlings purchased from the Chinese Academy of Forestry were propagated. Seedlings without buds, measuring 1–1.5 cm in height, were selected and transferred to 1/2 MS medium (Qingdao Haibo Biological) for rooting. Cultures were maintained in a tissue culture room for 45–60 days under a 10 h/14 h light/dark cycle, a light intensity of 1500–5000 Lx, a temperature of 24–27 °C, and a relative humidity of 60–80% [70]. After rooting, seedlings with uniform growth were selected for inoculation with endophytic nitrogen-fixing bacteria.

#### 4.5.2. Preparation of Bacterial Inoculum

Bacterial strains were cultured in 30 mL of Jnfb liquid medium in centrifuge tubes at 28 °C and 180 rpm for 48 h until an OD_600_ of 0.4 was reached. Under sterile conditions, 500 μL of bacterial culture was applied to uniformly growing apple tissue culture seedlings using a spray method. Ten replicates were used for each treatment, and ten uninoculated seedlings served as a blank control. Seedlings were grown under the same conditions as described above, and growth was monitored. After 60 days, plant height, number of leaves, chlorophyll content, and root length were measured.

### 4.6. Assessment of Plant Growth-Promoting Traits of Apple Endophytic Nitrogen-Fixing Bacteria

In addition to nitrogen fixation ability, other potential growth-promoting traits were also assessed using plate assays.

#### 4.6.1. IAA Production

IAA production was determined using the Salkowski colorimetric method [71]. Test strains were activated in an NA liquid medium, and 200 μL of the activated culture was inoculated into King’s B medium. After 5 days of incubation, cultures were centrifuged at 10,000 rpm for 5 min, and the supernatant was collected. One milliliter of the supernatant was mixed with an equal volume of Salkowski reagent and incubated in the dark for 30 min. The absorbance at OD530 was measured using a spectrophotometer.

#### 4.6.2. Phosphorus Solubilization

The phosphorus solubilization ability of the strains was preliminarily screened using the clear zone method on PKO inorganic phosphorus medium and Mengjinna organic phosphorus medium [72]. A single colony was picked with an inoculation loop and streaked onto a PKO solid medium. After incubation at 28 °C for 8 days, the presence or absence of a clear zone was observed. Strains exhibiting a clear zone were considered to have phosphorus solubilization ability and were further subjected to quantitative analysis. Quantitative determination of phosphorus solubilization was performed using the molybdenum blue colorimetric method [73]. A 200 μL aliquot of bacterial culture at OD_600_ = 0.2 was inoculated into test tubes containing 10 mL of PKO inorganic phosphorus medium and Mengjinna organic phosphorus medium, with three replicates per treatment. Cultures were incubated at 28 °C and 180 rpm for 5 days. One milliliter of the culture was transferred to a 2 mL centrifuge tube and centrifuged at 12,000 rpm for 10 min. The supernatant was transferred to a cuvette, and 1200 μL of molybdenum blue reagent was added. The volume was adjusted to 3 mL with ddH_2_O, and the mixture was allowed to stand for 30 min at room temperature. The absorbance at OD_700_ was measured using a UV spectrophotometer, and the amount of phosphorus solubilized was calculated using a standard curve.

#### 4.6.3. Siderophore Production

Siderophore production was determined using the Chrome Azurol S (CAS) assay [74]. Test strains were streaked onto an MSA-CAS medium and incubated at 28 °C. The formation of an orange-yellow halo around the colonies, indicative of siderophore production, was observed.

#### 4.6.4. Potassium Solubilization

Potassium solubilization was assessed using a spectrophotometric method [75] on a potassium feldspar medium. Test strains were streaked onto potassium feldspar medium plates and incubated at 28 °C. The formation of a clear zone around the colonies was observed as an indicator of potassium solubilization.

### 4.7. Statistical Analysis

For data analysis, Microsoft Excel was used for basic data organization, statistical calculations, and preliminary chart plotting, while Origin was employed for advanced visualization and in-depth analysis of complex data. In statistical analysis, alpha diversity indices were evaluated using non-parametric tests (such as the Kruskal–Wallis test) to determine whether significant differences existed among groups. If significant differences were found, Dunn’s test was subsequently applied for multiple comparisons. For Principal Coordinates Analysis (PCoA), Adonis analysis (permutational multivariate analysis of variance) was used to assess whether species compositions differed significantly between groups. For other continuous data, if normality and homogeneity of variance assumptions were met, a one-way analysis of variance (One-Way ANOVA) was used to determine group differences, followed by Tukey’s test for post hoc comparisons. If data failed to meet the assumptions of normality or equal variances, non-parametric tests were applied. All statistical analyses were performed using SPSS software version 19.0.

## 5. Conclusions

This study systematically revealed the distribution and composition of endophytic nitrogen-fixing bacteria (ENFB) in different tissues of apple trees, including roots, stems, and leaves. It identified distinct differences in diversity and dominant genera among various plant parts. A total of 138 ENFB strains were successfully isolated, and their potential plant growth-promoting traits—such as nitrogenase activity, indole-3-acetic acid (IAA) production, and siderophore secretion—were comprehensively evaluated. Furthermore, six representative ENFB strains and their combinations were shown to significantly enhance the growth of apple seedlings under in vitro conditions.

These findings not only enrich our understanding of the diversity and ecological roles of ENFB in apple trees but also provide a valuable microbial resource and solid theoretical foundation for the development of biofertilizers. The promising plant growth-promoting potential of these strains merits further validation under field conditions, paving the way for environmentally friendly and sustainable apple production strategies.

## Figures and Tables

**Figure 1 plants-14-01235-f001:**
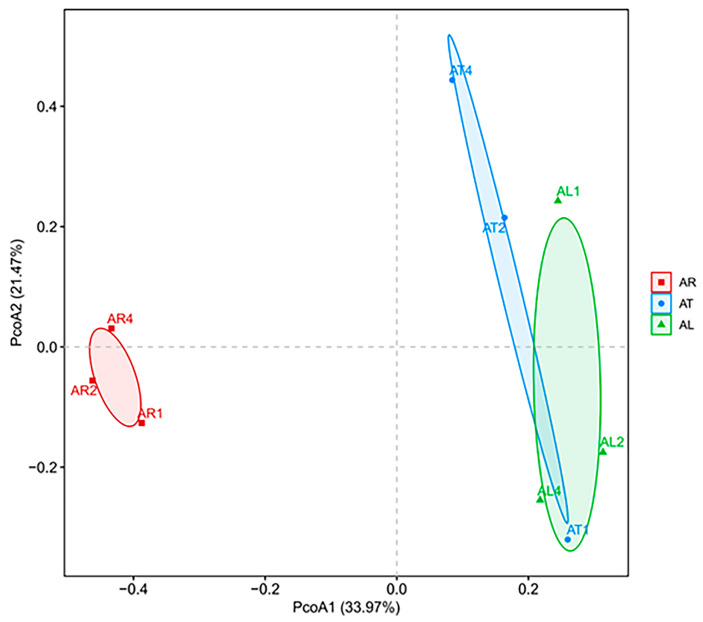
PcoA analysis of endophytic nitrogen-fixing bacteria community in different tissues of apple.

**Figure 2 plants-14-01235-f002:**
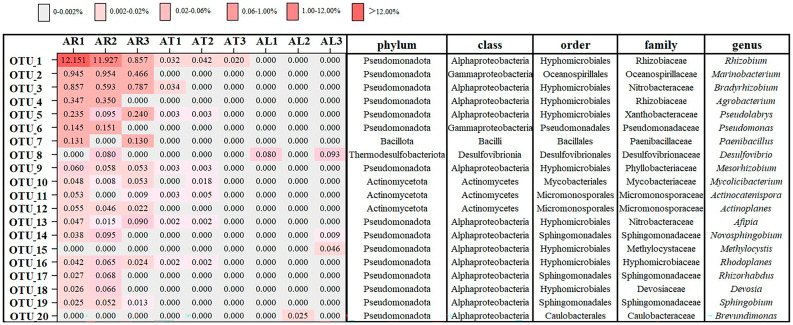
Relative abundance of the 20 most abundant OTUs in apple root, twig, and leaf samples.

**Figure 3 plants-14-01235-f003:**
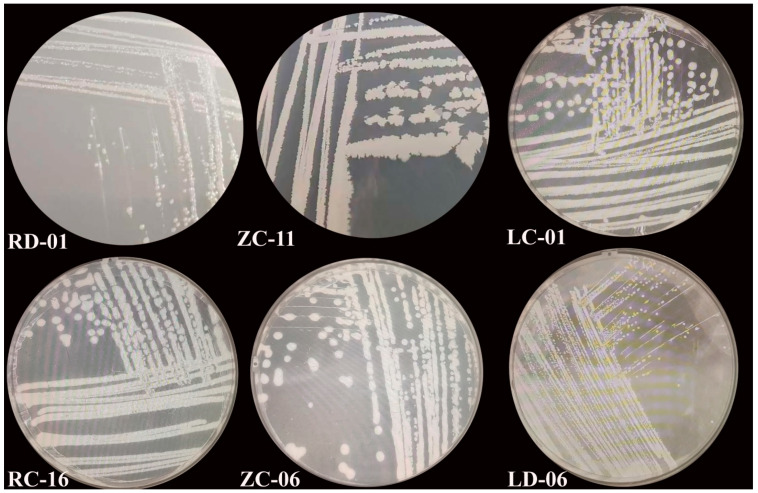
Colony morphology of partial endophytes from apple on Ash medium.

**Figure 4 plants-14-01235-f004:**
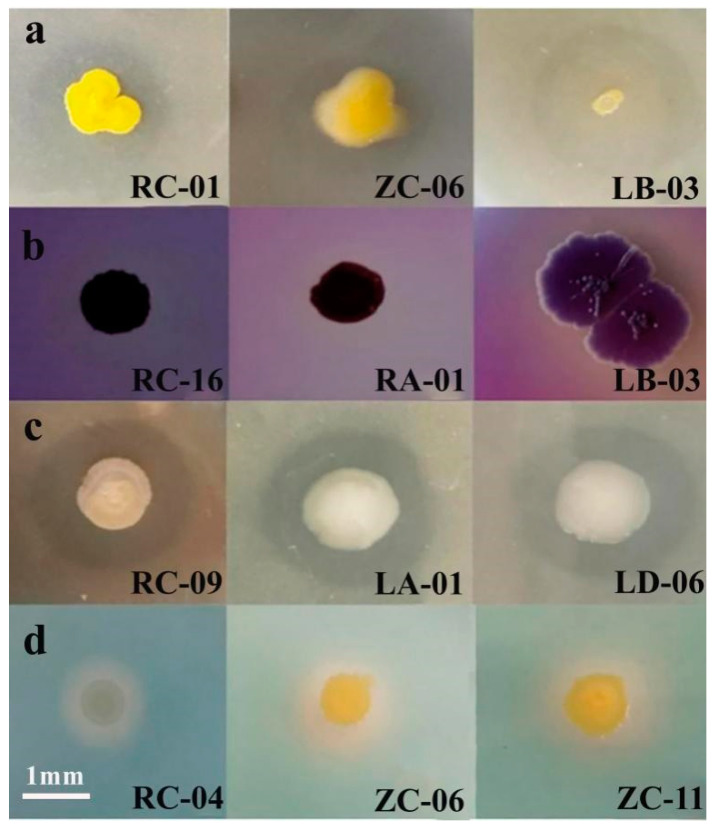
Solubilization of organic phosphorus (**a**), inorganic phosphorus (**b**), potassium (**c**), and siderophore production (**d**) by selected strains of endophytic nitrogen-fixing bacteria isolated from apple.

**Figure 5 plants-14-01235-f005:**
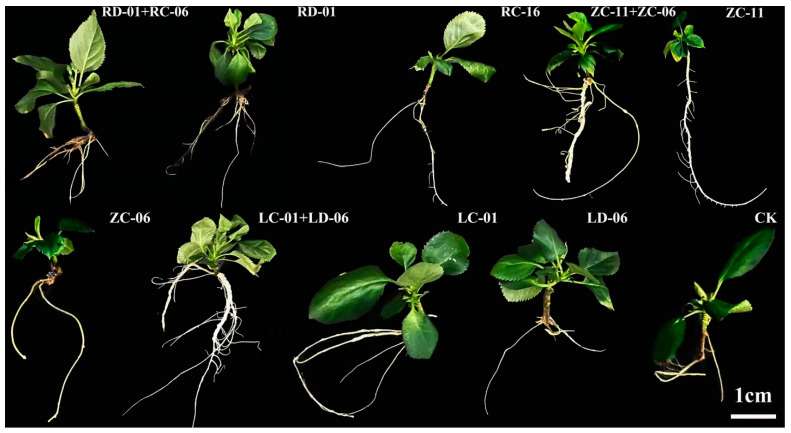
Grow-promoting effect by endophytic nitrogen-fixing bacteria from apple tree.

**Table 1 plants-14-01235-t001:** Diversity indices of endophytic nitrogen-fixing bacteria in different tissues of apple.

Tissue	Shannon Index Mean ± SE	Simpson Index	Chao Index	Ace Index	Goods Coverage
Root	4.98 ± 0.49 A	0.95 ± 0.02 A	1582.93 ± 278.21 a	1585.42 a	1.00
Twig	3.29 ± 0.4 AB	0.90 ± 0.04 AB	508.01 ± 417.94 ab	756.23 ± 775.31 ab	0.98
Leaf	2.25 ± 0.85 B	0.71 ± 0.26 B	100.6 ± 24.23 b	100.96 ± 23.98 b	1.00

Note: Values represent mean ± SD. Different uppercase letters indicate extremely significant differences (*p* < 0.01), and different lowercase letters indicate significant differences (*p* < 0.05).

**Table 2 plants-14-01235-t002:** The identification of endophytic nitrogen-fixing bacteria in apple tree.

	Isolate	Class	Closest Match (Biocloud)	Accession Number	Query Coverage (%)	Tissue
1	RA-07	Pseudomonadota	*Agrobacterium deltaense*	YTC 4121	99.68	R, L
2	RC-15	Pseudomonadota	*A. pusense*	LMG 25623	99.85	R, L
3	RC-14	Pseudomonadota	*A. radiobacter*	ATCC 19358	99.93	R, L
4	RC-09	Pseudomonadota	*Rhizobium-*AKKA_s	CF122	99.13	R
5	RB-08	Pseudomonadota	*Ancylobacter defluvii*	SK15	99.61	R
6	LD-05	Pseudomonadota	*Brucella anthropi*	ATCC 49188	99.78	L
7	LA-01	Pseudomonadota	*B. ciceri*	Ca-34	99.65	L
8	ZD-02	Pseudomonadota	*Methylobacterium brachiatum*	B0021	99.64	Z
9	ZC-08	Pseudomonadota	*M. goesingense*	iEII3	99.18	Z
10	ZC-10	Pseudomonadota	*M. komagatae*	002-079	96.86	Z
11	RD-01	Pseudomonadota	*Novosphingobium barchaimii*	LL02	99.06	R
12	RC-16	Pseudomonadota	*N. resinovorum*	NCIMB 8767	99.88	R, Z
13	ZC-04	Pseudomonadota	*N.-*SODQ_s	PhB55	99.13	R, Z
14	LD-10	Pseudomonadota	*Pseudomonas coleopterorum*	Esc2Am	99.65	L
15	LB-03	Pseudomonadota	*P. graminis*	DSM 11363	98.81	L
16	LD-07	Pseudomonadota	*P. putida*	EU348741.1	99.02	L
17	RB-02	Pseudomonadota	*Pseudorhodoplanes sinuspersici*	RIPI 110	99.49	R
18	RC-04	Pseudomonadota	*Sphingobium aquiterrae*	SKLS-A10	97.17	R
19	RA-01	Pseudomonadota	*S. scionense*	WP01	99.7	R
20	LD-09	Pseudomonadota	*S. yanoikuyae*	ATCC 51230	99.7	L
21	ZC-03	Pseudomonadota	*Sphingomonas taxi*	ATCC 55669	99.12	Z
22	LD-06	Pseudomonadota	*S. yabuuchiae*	GTC 868	100	L
23	ZC-11	Pseudomonadota	*S. yunnanensis*	YIM 003	99.03	Z
24	RA-02	Pseudomonadota	*Ancylobacter novellus*	DSM 506	99.33	R
25	ZC-01	Bacillota	*Bacillus velezensis*	CR-502	99.14	Z
26	ZC-12	Bacillota	*B. zanthoxyli*	1433	99.65	Z
27	ZD-05	Deinococcota	*Deinococcus depolymerans*	TDMA-24	98.91	Z
28	LC-01	Bacteroidota	*Sphingobacterium multivorum*	IAM 14316	99.67	L
29	LD-02	Actinomycetota	*Curtobacterium flaccumfaciens*	LMG 3645	99.88	L
30	LD-11	Actinomycetota	*Microbacterium aurum*	KACC 15219	98.25	L
31	LB-01	Actinomycetota	*Nocardioides phosphati*	WYH11-7	97.52	L
32	ZC-06	Actinomycetota	*Rhodococcus qingshengii*	JCM 15477	99.72	Z

**Table 3 plants-14-01235-t003:** Nitrogenase activity of endophytic nitrogen-fixing bacteria in apple tree.

Strain	Nitrogenase Activity (nmol/h·mL)	Strain	Nitrogenase Activity (nmol/h·mL)	Strain	Nitrogenase Activity (nmol/h·mL)
**RD-01**	**425.3 ± 13.69 a**	**ZC-06**	**982.28 ± 11.52 a**	**LC-01**	**543.95 ± 27.53 a**
RC-15	399.91 ± 10.48 a	ZC-11	951.91 ± 20.30 a	LD-10	513.43 ± 8.52 a
RA-07	228.03 ± 11.2 b	ZD-04	524.9 ± 6.32 b	LD-09	512.82 ± 50.36 ab
RC-14	205.66 ± 5.31 b	ZD-02	318.61 ± 2.51 c	LB-03	473.63 ± 5.21 b
RA-01	166.26 ± 4.32 bc	ZC-08	283.45 ± 6.21 cd	LD-02	428.23 ± 13.69 c
RB-08	123.77 ± 6.02 c	ZC-12	227.05 ± 4.28 d	LB-01	420.66 ± 7.30 c
RC-09	121.1 ± 3.10 c	ZC-10	189.7 ± 1.52 de	LA-01	412.6 ± 6.33 c
RB-02	74.95 ± 4.10 d	ZD-05	115.36 ± 4.52 e	LD-11	338.19 ± 25.41 d
RA-01	48.58 ± 1.94 de	ZC-03	104.36 ± 2.64 e	LD-07	308.35 ± 5.74 d
RC-04	47.12 ± 1.22 e	ZC-01	74.95 ± 4.25 ef	LD-05	44.68 ± 5.9 f
RC-16	27.1 ± 0.15 f			LD-06	26.86 ± 4.25 f

Note: In the strain numbers, “R” represents strains derived from roots, “L” represents strains derived from leaves, and “Z” represents strains derived from twigs. Values represent mean ± SD. Different lowercase letters indicate a significant difference (*p* < 0.05).

**Table 4 plants-14-01235-t004:** Growth-promoting effects of endophytic nitrogen-fixing bacteria on apple tissue culture seedlings.

Treatment	Height (cm)	Leaf Number	Chlorophyll Content (mg/g)	Root Length (cm)
RD-01	4.4 ± 0.2 b	7 ± 3.46 a	16.43 ± 0.27 b	2.9 ± 0.17 d
RC-16	3.3 ± 0.1 c	3.33 ± 0.58 c	10.21 ± 0.22 cd	2.2 ± 0.1 d
RD-01 + RC-16	5.3 ± 0.26 a	6.33 ± 1.15 ab	21.34 ± 0.43 a	2.1 ± 0.17 d
ZC-11	1.73 ± 0.06 e	5 ± 1 bc	6.03 ± 0.02 e	7.5 ± 0.46 a
ZC-06	1.93 ± 0.23 e	3.33 ± 0.58 c	3.59 ± 0.18 f	5.8 ± 0.35 b
ZC-11 + ZC-06	2.07 ± 0.49 de	6.33 ± 0.58 ab	9.03 ± 0.16 d	5 ± 0.17 b
LC-01	3.07 ± 0.76 c	4 ± 0 bc	11.24 ± 0.35 c	4 ± 0.46 c
LD-06	2.8 ± 0.3 cd	5.33 ± 0.58 abc	14.75 ± 0.65 b	2.13 ± 0.15 d
LC-01 + LD-06	2.07 ± 0.25 de	6 ± 0 ab	20.01 ± 1.76 a	5.4 ± 0.35 b
CK	2.17 ± 0.83 de	3.33 ± 0.58 c	5.13 ± 2.65 ef	2.23 ± 1.25 d

Note: Values represent mean ± SD. Different lowercase letters indicate significant differences (*p* < 0.05) between the single and combined inoculations and the control. CK represents the uninoculated tissue culture seedling control.

**Table 5 plants-14-01235-t005:** Four types of nitrogen-free media used in this study.

Ingredients (g/L)	Jensen	A4	Ash	Jnfb
C_12_H_22_O_11_	20.0	20.0	-	-
K_2_HPO_4_3H_2_O	1.31	-	-	0.785
NaCl	0.5	-	0.2	0.1
CaCO_3_	2.0	0.1	5.0	-
FeSO_4_·7H_2_O	0.18	-	-	-
MgSO_4_·7H_2_O	1.0	-	-	-
Na_2_MnO_4_	0.005	-	-	-
MgSO_4_7H_2_O	-	0.5	0.2	0.2
Na_2_HPO_4_12H_2_O	-	5.0	-	-
FeCl_3_	-	0.005	-	-
Mannitol	-	-	10.0	-
KH_2_PO_4_	-	-	0.2	-
CaSO_4_2H_2_O	-	-	0.1	-
C_4_H_6_O_5_	-	-	-	5.0
CaCl_2_	-	-	-	0.02
Fe_3_-EDTA	-	-	-	0.066
Biotin	-	-	-	1.0
VB_6_	-	-	-	0.1
KOH	-	-	-	4.5
Na_2_MoO_4_2H_2_O	-	-	-	0.2
MnSO_4_H_2_O	-	-	-	0.235
H_3_BO_3_	-	-	-	0.28
CuSO_4_5H_2_O	-	-	-	0.008
ZnSO_4_7H_2_O	-	-	-	0.024
Agar	20.0	20.0	20.0	20.0

**Table 6 plants-14-01235-t006:** Selected strains for growth promotion assay.

Source Tissue	High Nitrogenase Activity Strain	Plant Growth-Promoting Strain	Consortium
Root	RD-01	RC-16	RD01 + RC16
Twig	ZC-06	ZC-11	ZC11 + ZC06
Leaf	LC-01	LD-06	LC01 + LD06

## Data Availability

All raw sequence data reported in this paper have been deposited in the Genome Sequence Read Archive in the National Genomics Data Center, China National Center. All data can be viewed at https://www.ncbi.nlm.nih.gov/datasets/taxonomy/tree/ (accessed on 22 September 2024) and http://eztaxon-e.ezbiocloud.net (accessed on 24 September 2024) downloaded through the weblink.

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
