# Peer review of "Genetic Diversity and Growth-Promoting Functions of Endophytic Nitrogen-Fixing Bacteria in Apple"

_plants, 2025, doi:10.3390/plants14081235_

Round 1
Reviewer 1 Report
Comments and Suggestions for Authors
Presented manuscript provides a novel data on plant growth promotion bacteria associated with apple threes.
I found the manuscript interesting, novel and well performed.
However several minor revisions should be done before publishing:
Phyla names should be corrected being corresponds to valid names throughout all the manuscript doi: https://doi.org/10.1099/ijsem.0.005056
Introduction section should be supplemented with data on plant growth promotion consortia of other fruit trees, biodiversity and support mechanisms of BGPB briefly.
line 370 - "citation needed" :) Should be corrected
Author Response
Reviewer 1
Presented manuscript provides a novel data on plant growth promotion bacteria associated with apple threes.I found the manuscript interesting, novel and well performed. However several minor revisions should be done before publishing:
Phyla names should be corrected being corresponds to valid names throughout all the manuscript doi: https://doi.org/10.1099/ijsem.0.005056
Response: Thank you for the suggestion. We have revised the aforementioned phylum names throughout the manuscript and have cited the relevant literature (No. 66) in section 4.3.
Introduction section should be supplemented with data on plant growth promotion consortia of other fruit trees, biodiversity and support mechanisms of BGPB briefly.
Response: Thank you for your suggestions. We have rewritten the Introduction section as recommended, merging related data and progress. References No. 3, 6, 7, 8, 16, 21, 22, and 24 have been incorporated into the revised text and are included in the References section.
line 370 - "citation needed" :) Should be corrected
Response: Thank you for your meticulous proofreading. Reference No. 63 has been added to the text at this point.
Reviewer 2 Report
Comments and Suggestions for Authors
Dear Authors
This study presents interesting results on the diversity and potential for promoting the growth of endophytic nitrogen fixing bacteria in apple trees
The experiments showed that certain bacteria and combinations thereof significantly enhanced the growth of young apples in tissue culture. The study highlights the potential of these endophytic bacteria to develop sustainable fertilizers in apple cultivation.
I have some comments:
Abstract: too long. limit the length and focus on the aims and results
The introduction is short. Add some information. There could be a more direct link between the problems caused by excessive nitrogen fertilizer use in apple trees and the specific objective of the study to explore endophytic nitrogen fixing bacteria as an alternative. Rewording of the last paragraph of the introduction to emphasise more emphatically how investigating the diversity and functions of endophytic nitrogen fixing bacteria in apple trees is a direct response to the environmental and economic problems associated with the use of chemical fertilisers
L348 write what supposed to be the “red soil”
L370 [Citation needed) something has escaped your attention
2.1.3, in the description of the results at the genus level, the relative abundance of the top 10 genera is reported in the root and travel samples, but for leaves all 6 genera are reported with their relative abundance. It would be better to also report the top few genera in leaves or explain why all genera are reported for leaves.
2.2, reference is made to "32 taxonomic groups (Table 2)", but Table 2 presents 32 strains, each with its taxonomic identity. Clarify whether the 32 taxonomic groups correspond to the 32 strains presented or whether some groups include multiple strains. The wording should be more precise.
2.5 describes the inoculation of in vitro grown apple seedlings. Although the results are encouraging, transferring these results to field conditions requires caution. Microenvironmental conditions and interaction with other microbes in the soil may affect the efficacy of the inoculated bacteria.
4.7 Statistical Analysis", it would be better to indicate more specifically the statistical tests used for each type of data. In addition, when presenting the results in the tables and in the text, standard deviations or standard errors could be reported more precisely (e.g., reported only as '±').
In the discussion, the need for future studies under field conditions to evaluate the actual effectiveness of the selected strains and combinations in promoting apple growth could be emphasized
Figure 2 (Relative abundance of the 20 most abundant OTUs): the image is somewhat crowded, especially the labels on the OTU axis are difficult to read.
Table 3, Table 4: The use of small lowercase letters is reported to indicate statistically significant differences (P < 0.05). Insufficient detail is provided on the exact statistical tests used for between-group comparisons in the growth promotion experiments (e.g., which form of ANOVA was used and which post-hoc tests).
Comments on the Quality of English LanguageEnglish language is fine but authors can se clearer transitional phrases between paragraphs to ensure a more logical and smooth flow of ideas
Author Response
Reviewer 2
Dear Authors
This study presents interesting results on the diversity and potential for promoting the growth of endophytic nitrogen fixing bacteria in apple trees
The experiments showed that certain bacteria and combinations thereof significantly enhanced the growth of young apples in tissue culture. The study highlights the potential of these endophytic bacteria to develop sustainable fertilizers in apple cultivation.
I have some comments:
Abstract: too long. limit the length and focus on the aims and results
Response: Thank you for your comments. The Abstract section has been completely rewritten to emphasize the aims and key results, aligning it with the journal's style.
The introduction is short. Add some information. There could be a more direct link between the problems caused by excessive nitrogen fertilizer use in apple trees and the specific objective of the study to explore endophytic nitrogen fixing bacteria as an alternative. Rewording of the last paragraph of the introduction to emphasise more emphatically how investigating the diversity and functions of endophytic nitrogen fixing bacteria in apple trees is a direct response to the environmental and economic problems associated with the use of chemical fertilisers.
Response: As per Reviewer 1's suggestion, the Introduction section has also been rewritten and improved.
L348 write what supposed to be the “red soil”
Response: The soil traits – low soil pH, low organic matter content, poor nutrient retention capacity, and proneness to compaction – have been added after 'red soil'.
L370 [Citation needed) something has escaped your attention
Response: We appreciate your meticulous proof check. Here, a reference No.63 was added in the text.
2.1.3, in the description of the results at the genus level, the relative abundance of the top 10 genera is reported in the root and travel samples, but for leaves all 6 genera are reported with their relative abundance. It would be better to also report the top few genera in leaves or explain why all genera are reported for leaves.
Response: Thank you for your meticulous attention on this point. In this study, the number of genera from roots is more than from leaves. All 6 genera from leaves were descripted in the main text. The reason why in the leaves was lower, we don’t have scientific evidence to explain this in the study. Therefore, we just present a fact on it.
2.2, reference is made to "32 taxonomic groups (Table 2)", but Table 2 presents 32 strains, each with its taxonomic identity. Clarify whether the 32 taxonomic groups correspond to the 32 strains presented or whether some groups include multiple strains. The wording should be more precise.
Response: The 138 isolated strains were identified and classified into 32 distinct taxonomic groups, with some groups containing multiple strains. To facilitate subsequent experiments, one representative strain was selected from each taxonomic group; this selection process is detailed in section 2.4 of the manuscript. Table 2 presents the identification results and specific taxonomic affiliations of these 32 representative strains.
2.5 describes the inoculation of in vitro grown apple seedlings. Although the results are encouraging, transferring these results to field conditions requires caution. Microenvironmental conditions and interaction with other microbes in the soil may affect the efficacy of the inoculated bacteria.
Response: Thank you for the suggestions. We strongly agree with the reviewers' comments. Indeed, this paper only presents preliminary in vitro experiments, and further field screening trials will be necessary before any large-scale field application can be considered. This point has been addressed in the corresponding location within the Discussion section of the main text, further clarifying the limitations of the in vitro experiments and the importance of subsequent field trials.
4.7 Statistical Analysis", it would be better to indicate more specifically the statistical tests used for each type of data. In addition, when presenting the results in the tables and in the text, standard deviations or standard errors could be reported more precisely (e.g., reported only as '±').
Response: Thank you for your suggestion. We have refined the description of the statistical analysis methods in section 4.7 (Methods) of the manuscript and have specified in the table footnotes that the data are presented as 'mean ± standard deviation
In the discussion, the need for future studies under field conditions to evaluate the actual effectiveness of the selected strains and combinations in promoting apple growth could be emphasized
Response: We greatly appreciate the reviewers' suggestion. Indeed, this study presents preliminary in vitro experiments, and further field screening trials will be essential for potential large-scale field applications. This point has been discussed in the relevant section of the Discussion within the main text, further clarifying the limitations of the in vitro experiments and underscoring the importance of subsequent field trials
Figure 2 (Relative abundance of the 20 most abundant OTUs): the image is somewhat crowded, especially the labels on the OTU axis are difficult to read.
Response: Figure 2 has been adjusted with respect to font size and table row height.
Table 3, Table 4: The use of small lowercase letters is reported to indicate statistically significant differences (P < 0.05). Insufficient detail is provided on the exact statistical tests used for between-group comparisons in the growth promotion experiments (e.g., which form of ANOVA was used and which post-hoc tests).
Response: The variance analysis of the data in the manuscript was performed using One-Way ANOVA to determine differences between groups, followed by Tukey's HSD post-hoc test for multiple comparisons. If the data did not meet the assumptions of normality or homogeneity of variances, non-parametric tests were employed. Specific details have been extensively supplemented in the Methods section, part 4.7, of the manuscript.
English language is fine but authors can se clearer transitional phrases between paragraphs to ensure a more logical and smooth flow of ideas
Response: We checked the entire manuscript and revised those could be cause abrupt transfer and links between paragraphs.
Reviewer 3 Report
Comments and Suggestions for Authors
Dear Author,
After reviewing your manuscript "Genetic Diversity and Growth-Promoting Functions of Endophytic Nitrogen-Fixing Bacteria in Apple," I found that it effectively demonstrates the genetic diversity and growth-promoting functions of endophytic nitrogen-fixing bacteria in apple trees using high-throughput sequencing and inoculation assays to evaluate their potential as biofertilizers. However, I suggest the following improvements:
-
Improve the clarity of the summary conclusion so that it clearly demonstrates the study’s contribution and novelty.
-
In the introduction, clearly state the justification for the study by including updated data or comparisons with recent research on the environmental and economic impacts of fertilizer use.
-
In the methodology, provide more detailed descriptions of the methods, include proper citations for the primers used, organize the composition of the culture media in tables or subsections for clarity, and elaborate further on the statistical analysis employed.
-
In the results section, present the data with more thorough analysis and interpretation in the text. Ensure that figures and tables are clearly referenced and explained, and explicitly link numerical results (such as diversity indices and nitrogenase activity) to your hypothesis.
-
In the discussion, provide additional bibliographic support for assertions regarding genera like Rhizobium and Desulfovibrio. Compare your findings with similar studies, highlighting both similarities and differences, and support your claims with more references and experimental data. Also, discuss the limitations of your study and propose future research directions.
-
In the conclusion, concisely summarize the main contributions without repeating data already presented.
-
Verify and standardize the formatting of your references to ensure consistency with the target journal’s guidelines.
no comments
Author Response
Reviewer 3
Dear Author,
After reviewing your manuscript "Genetic Diversity and Growth-Promoting Functions of Endophytic Nitrogen-Fixing Bacteria in Apple," I found that it effectively demonstrates the genetic diversity and growth-promoting functions of endophytic nitrogen-fixing bacteria in apple trees using high-throughput sequencing and inoculation assays to evaluate their potential as biofertilizers. However, I suggest the following improvements:
- Improve the clarity of the summary conclusion so that it clearly demonstrates the study’s contribution and novelty.
Response:Thank you for the reviewers' suggestion. Section 5, Conclusion, of the manuscript has been rewritten based on this feedback.
- In the introduction, clearly state the justification for the study by including updated data or comparisons with recent research on the environmental and economic impacts of fertilizer use.
Thank you for the reviewers' suggestion. Relevant content has been added to section 1, Introduction.
- In the methodology, provide more detailed descriptions of the methods, include proper citations for the primers used, organize the composition of the culture media in tables or subsections for clarity, and elaborate further on the statistical analysis employed.
Response: Thank you for the reviewers' suggestion. The statistical analysis methods used for the data analysis in the manuscript have been further detailed in section 4.7 (Methods). The references for all primer sequences have been included [65], and the culture medium recipes have been presented in tabular form (Table 5).
- In the results section, present the data with more thorough analysis and interpretation in the text. Ensure that figures and tables are clearly referenced and explained, and explicitly link numerical results (such as diversity indices and nitrogenase activity) to your hypothesis.
Response: Thank you for the reviewers' suggestion. All figures and tables in the Results section, including secondary labels within figures (e.g., a, b, c), have been explained and explicitly linked in the text. The data within the tables have been described and interpreted. Corresponding explanations have been added to Tables 1, 3, and 4.
- In the discussion, provide additional bibliographic support for assertions regarding genera like Rhizobium and Desulfovibrio. Compare your findings with similar studies, highlighting both similarities and differences, and support your claims with more references and experimental data. Also, discuss the limitations of your study and propose future research directions.
Response: Thank you for the reviewers' suggestion. Supporting literature for the related discussion has been added to section 4, Discussion. Similarities and differences have also been compared. References [34] and [35] have been added to section 3.2.
- In the conclusion, concisely summarize the main contributions without repeating data already presented.
Response: Thank you for the reviewers' suggestion. Section 5, Conclusion, has been rewritten.
- Verify and standardize the formatting of your references to ensure consistency with the target journal’s guidelines.
Response: Thank you for the reviewers' suggestion. The References section has been checked and revised.
Round 2
Reviewer 2 Report
Comments and Suggestions for Authors
Dear Authors
The article text has been greatly upgraded, in every respect, introduction, discussion, conclusions, statistical analysis, tables etc.
Comments on the Quality of English Language
The English is fine but it could be improved